# Genome-Wide Association Mapping of Processing Quality Traits in Common Wheat (*Triticum aestivum* L.)

**DOI:** 10.3390/genes14091816

**Published:** 2023-09-18

**Authors:** Hui Jin, Yuanyuan Tian, Yan Zhang, Rui Zhang, Haibin Zhao, Xue Yang, Xizhang Song, Yordan Dimitrov, Yu-e Wu, Qiang Gao, Jindong Liu, Jumei Zhang, Zhonghu He

**Affiliations:** 1Institute of Forage and Grassland Sciences, Heilongjiang Academy of Agricultural Sciences, Harbin 150086, China; jinhuicaas@126.com (H.J.); yxflax@126.com (X.Y.); h.song@foxmail.com (X.S.); ydtsvetkov@163.com (Y.D.); lunawye@163.com (Y.W.); 2National Wheat Improvement Center, Institute of Crop Sciences, Chinese Academy of Agricultural Sciences, Beijing 100000, China; tianyuanyuan0302@163.com (Y.T.); zhangyan07@caas.cn (Y.Z.); liujindong@caas.cn (J.L.); 3Horticultural Branch of Heilongjiang Academy of Agricultural Sciences, Harbin 150086, China; 18500654770@163.com

**Keywords:** bread wheat, GWAS, mixograph, SNP array, wheat quality

## Abstract

Processing quality is an important economic wheat trait. The marker-assisted selection (MAS) method plays a vital role in accelerating genetic improvement of processing quality. In the present study, processing quality in a panel of 165 cultivars grown in four environments was evaluated by mixograph. An association mapping analysis using 90 K and 660 K single nucleotide polymorphism (SNP) arrays identified 24 loci in chromosomes 1A, 1B (4), 1D, 2A, 2B (2), 3A, 3B, 3D (2), 4A (3), 4B, 5D (2), 6A, 7B (2) and 7D (2), explaining 10.2–42.5% of the phenotypic variances. Totally, 15 loci were stably detected in two or more environments. Nine loci coincided with known genes or QTL, whereas the other fifteen were novel loci. Seven candidate genes encoded 3-ketoacyl-CoA synthase, lipoxygenase, pyridoxal phosphate-dependent decarboxylase, sucrose synthase 3 and a plant lipid transfer protein/Par allergen. SNPs significantly associated with processing quality and accessions with more favorable alleles can be used for marker-assisted selection.

## 1. Introduction

Dough rheological properties with significant effects on end-use products can be evaluated by mixograph [1,2]. Midline peak time (MPT), midline 8 min band width (MTxW), mid-line peak width (MPW) and midline peak value (MPV) are related to processing quality [3,4]. Dough properties are highly variable among wheat cultivars [5,6] and processing quality is a major breeding objective in wheat breeding programs [7,8].

Dough rheological properties are quantitatively inherited and largely controlled by multiple minor genes [9,10,11,12]. It is difficult to evaluate dough rheological properties in traditional breeding because they cannot be measured in the early segregating generations of crosses due to limited quantities of seed. Moreover, measurement of mixograph-related traits requires a professional grain chemistry laboratory. Marker-assisted selection (MAS) could be a useful approach to improve wheat processing quality [7]. Dough strength is a typical quantitative trait controlled by multiple minor genes. A number of studies have been conducted which focus on identifying QTLs for mixographs and a series of QTLs for mixograph-related traits [2,13,14,15,16,17] were reported mainly by bi-parental linkage mapping. Barakat et al. [2] reported 108 loci for farinograph- and mixograph-related traits on all 21 wheat chromosomes in two double haploid (DH) populations. However, the short-coming of the method is that only two alleles at any single locus can be evaluated in each cross. Echeverry-Solarte et al. [17] reported 31 loci for mixograph-related traits by an RIL population developed from a cross of an elite wheat line (WCB414) and an exotic genotype with supernumerary spikelets, and each explained 3.2–41.2% phenotypic variations.

The results of those studies were mainly based on relatively low numbers of simple sequence repeat (SSR) or diversity array technology (DArT) markers [13,14] and were difficult to apply in gene cloning and MAS. The development of the wheat 90 K [18], 660 K [19] and 55 K SNP genotyping assays [19,20] has made it possible to genotype large populations with high-density SNPs. As a result, genome-wide association studies (GWAS) have been extensively conducted to explore the extant allelic diversity concerning numerous agronomic traits [21,22]. GWAS utilizes large amounts of markers distributed across the entire chromosomes of a species genome to find marker–trait relationships using LD as its basis, thereby uncovering significant positions. LD represents a naturally occurring phenomenon within a group during selection and evolution processes whereby nonrandom correlations occur among genes located at various sites within the same individual. If the probability of one specific allele existing at two distinct locations exceeds random expectations, then these two places exhibit LD. The process from unlinked imbalance to balanced linkage occurs throughout LD attenuation. Species differences exist regarding their distance of LD decay, with interspecific species such as maize and rapeseed exhibiting much farther distances than intraspecific ones such as wheat and rice. The level of LD also depends on factors including recombination, mutation, population structure, sample size, selective pressure, genetic drift, founder effects, admixture rates, etc. Among them, recombination and mutations play crucial roles in influencing LD levels [21,22,23].

To avoid false positives resulting due to any type of familiarity found amidst members comprising studied collectives alongside similar concerns about establishing appropriate mathematical formulas proves useful. General Linear Model (GLM), which was initially introduced, only considers impacts stemming from social arrangements, henceforth causing some degree of inaccurate outcomes because GLM does not take enough measures aimed towards controlling consequences originating via interactions caused jointly by kin and societal constructs. In contrast, MLM has been developed more recently since this methodology incorporates Q (population structure)-based assessments too—K (kinship); therefore, it better manages situations arising out of overlapping ramifications exerted either way by those concepts contributing toward finalized conclusions drawn downstream. At current times, researchers have implemented applicable usage of all sorts of techniques encompassing crop quantity features regularly utilizing mixed linier regression versions (MLM) [21,22,23].

Association analysis is a quantitative genetic analysis method based on LD among allelic variants at the same locus, using natural populations as research materials. By investigating the association between group genotype data and phenotypic data, target trait genes can be discovered. The advantage of this approach includes, firstly, that it does not require constructing biparental populations like linkage analysis but uses existing natural populations, high-generation crosses, local varieties and wild species instead to significantly shorten study cycles and improve work efficiency [21]. Secondly, its use of diverse sources for inheritance variation allows simultaneous detection of multiple alleles at the same position with increased potential applicability across different breeding backgrounds. Thirdly, because complexity exists in interspecific comparisons due to differences in trait expression patterns within various breeds or germplasm pools, one sample may serve several purposes during an assay. Fourthly, higher resolution occurs when utilizing naturally occurring recombination events that are more informative than artificial ones used by breeders. However, there also exist limitations such as low precision caused by limited diversity levels, which leads to reduced accuracy compared to other methods involving marker-assisted selection. Additionally, challenges include complicated sampling histories from variable geographic locations along with concerns about confounding effects related to family structure and environmentally driven influences upon gene frequencies. Furthermore, rare variant discovery suffers lower sensitivity rates, resulting in possible loss of important heritable variations. Combining both methods enables effective utilization of their respective strengths while addressing deficiencies encountered throughout each process, ultimately improving overall effectiveness regarding efficient identification of key hereditary factors underlying multifactorial characteristics [22,23]. Moreover, GWAS can be performed much faster and at lower cost because it bypasses the time of developing biparental populations [24,25]. GWAS has been used to conduct genetic analysis of a wide range of agronomic traits and resistance to diseases [25,26], grain processing and end-use quality [27,28], tolerance to abiotic stress [29], and yield-related traits [30,31].

The Yellow and Huai River Valleys Facultative Wheat Region (YHVFWR) is the largest wheat production region in China. Breeding cultivars with superior processing quality could be greatly enhanced using markers developed from single-nucleotide polymorphisms (SNPs). In this study, SNPs associated with processing quality in a panel of 165 elite wheat accessions mainly from the YHVFWR were used to (1) dissect the genetic architecture of mixograph-related traits, (2) identify SNPs significantly associated with mixograp-related traits and (3) search for candidate mixograph-related traits genes for further study.

## 2. Materials and Methods

### 2.1. Plant Materials and Field Trials

In the current investigation, a comprehensive collection consisting of 165 diversified varieties was utilized; specifically, they were comprised of 143 germplasms sourced from both the Yellow and Huai River Valley Facultative Wheat Region located across mainland China while additionally integrating another 22 samples originating from different nations, such as Italy (9); Argentina (7); Japan (4); Australia (1); along with one sample collected each from Turkish territory. These hexaploid wheats represented all types found worldwide based upon their geographical origin or agronomic performance. Furthermore, every single variety had already received permission before being stored into the national seed bank maintained by the Institute of Plant Industry affiliated with the Crop Science Society of China.

The 165 accessions used in GWAS to identify the loci for quality traits assessed by mixograph were grown at Suixi in Anhui province and Anyang in Henan in the 2012–2013 and 2013–2014 cropping seasons. Field trials were conducted in randomized complete blocks design (RCBD). Agronomic management followed local practices. Each plot contained three 2 m rows spaced 20 cm apart, and there were 3 replications.

### 2.2. Mixograph-Related Traits Evaluated and Statistical Analyses

Clean samples of at least 300 grains were tempered overnight to the 14%, 15% and 16% moisture contents normally used to mill soft, medium and hard types, respectively. All samples were milled at 60% flour extraction using a Brabender Quadrumat Junior Mill (Brabender Inc., Duisberg, Germany). Mixographs have several advantages, including small sample amounts (usually 10 g, minimum 2 g), high daily processing samples, simple operation and rich curve information. They are increasingly widely used in research on the rheological properties of dough due to their excellent performance characteristics. Mixograph measurements were taken with 10 g of flour per sample on a 14% moisture basis using the National Manufacturing Mixograph (National Manufacturing, TMCO Division, Lincoln, NE, USA), according to the AACC (2000) method 54-40A. BLUP across environments was analyzed using the PROCMIXED function in SAS v9.3.

### 2.3. Genotyping and Population Structure

Cultivars were genotyped by the 90 K SNP and 660 K SNP arrays by CapitalBio (Beijing, China). The SNP chip genotyping procedure involves six steps: (1) sample whole-genome amplification at approximately 1000 folds; (2) fragmentation treatment on the diluted PCR product; (3) hybridization between segmented DNAs and chips; (4) single base extension reactions at specific loci; (5) staining; and (6) scanning imaging. Due to common wheat’s hexaploid nature, we use combinations of GeneStudio v2011.1 and GeneStudio Polyploid Clustering V1.0 for genetic typing. First, raw image scans from gene expressions were read out through GeneStudio v2011.1; then, clustering based on ploidy level was performed via GeneStudio Polyploid Clustering V1.0. Filter criteria followed four standards: (1) eliminating markers without differences among parents; (2) homozygous alleles assumed missing data; (3) removing marker datasets where more than 10 percentage points of values have been removed; and (4) one particular variant site should meet either condition that either one or two types account for no greater than 0.7 or equal to or larger than 0.3, respectively.

The Chinese Spring (IWGSC v1.0) reference genome was used for GWAS. Population structure, principal components analysis (PCA), NJ-tree and LD decay analysis were reported in a previous study [25]. Population structure was assessed utilizing 2000 polymorphic SNP markers sourced from the 660 K SNP arrays. The analysis was conducted using Structure v2.3.4 (http://pritchardlab.stanford.edu/struc-ture.html) (accessed on 5 July 2022). For each K value ranging from 2 to 12, five independent runs were executed employing an admixture model. Each run consisted of 100,000 Markov Chain iterations that were recorded, preceded by 10,000 burn-in periods. To anticipate the actual count of subpopulations, an ad hoc quantity statistic denoted as ΔK, reliant on the rate of logarithmic probability alteration between consecutive K values, was employed [25]. Broad-sense heritability (*h_b_*^2^) of mixograph-related traits was calculated as *hb*^2^ = *σg*^2^/(*σg*^2^ + *σge*^2^/r +*σε*^2^/re), where *σge*^2^, *σε*^2^ and *σg*^2^ mean the genotype × environment interaction, residual error variances and genotype, respectively. Of these, e and r were the number of environments and the number of replicates per environment, respectively.

### 2.4. Association Mapping and the Identification of Candidate Genes

The mixed linear model (MLM, PCA + K) was used to avoid the spurious marker–trait associations (MTAs) by Tassel v5.0 [27]. Both the kinship matrix and PCA were estimated by Tassel v5.0. Markers with an adjusted −log10 (*p*-value) ≥ 3.0 were regarded as MTAs because Bonferroni–Holm correction was too conservative. Manhattan and Q-Q plots were drawn by the CMplot (R 3.6.5).

Candidate genes associated with loci consistently identified across two or more environments were pinpointed. The ensuing procedures were undertaken to ascertain candidate genes for noteworthy or steadfast quantitative trait loci (QTL). Firstly, a thorough search was conducted to retrieve all genes located within the linkage disequilibrium (LD) block vicinity surrounding the peak single-nucleotide polymorphism (SNP) (within a ± 3.0 Mb range, based on prior LD decay analysis) of each significant QTL from the IWGSC V1.0 dataset. Subsequently, all accessible SNPs located within these genes were scrutinized. Genes (excluding those encoding hypothetical proteins, transposon proteins and retrotransposon proteins) harboring SNPs within coding regions, with the potential to induce missense mutations, were designated as candidate genes. Given the substantial regulation of processing quality traits by diverse phytohormones, as well as factors such as glycolysis, signal transduction and cell growth, genes participating in these pathways were classified as high-confidence candidate genes for processing quality traits. Flanking sequences of significantly associated SNPs (including the LD decay interval of peak markers around 3.0 Mb) were used in BLASTx against the NCBI database and reference genome annotations from IWGSC v1.0 was used to predict candidate genes.

## 3. Results

### 3.1. Genotyping and Population Structure Analysis

After filtering, 259,922 SNPs were used in GWAS of mixograph-related traits [25]. Population structure, neighbor-joining (NJ) tree and PCA analysis identified three subgroups of accessions [25]; Subgroup I mainly originated from Shandong; Subgroup II included cultivars mainly from Henan, Anhui and Shaanxi; and most Subgroup III cultivars were from Henan (Appendix A) [25]. LD decay for the whole genome was about 8 Mb, with the D genome 11 Mb, the A genome 6 Mb and the B genome 4 Mb (Appendix A) [25]. The SNP density across the whole genome was about 18.5 SNPs/Mb [25].

### 3.2. Phenotypic Evaluation

All mixograph-related traits showed continuous variation (Appendix A; Appendix A). The mean values of MPT, MPV, MPW and MTxW were 3.16 (1.52–6.49), 47.84 (36.3–63.76), 18.32 (10.19–26.75) and 6.29 (2.5–17.67), respectively. ANOVA revealed significant effects (*p* <0.01) of genotypes, environments and genotype × environment interactions on each processing-related quality trait (Table 1). Broad-sense heritability (*h*^2^) estimated for MPW, MPV, MPT and MTxW was 0.75, 0.72, 0.69 and 0.71, respectively. MPT and MPW showed significant (*p* < 0.01) and positive correlations with MTxW (r = 0.885 and 0.448), and the MPV showed a significant (*p* < 0.01) and positive correlation with MPW (r = 0.867).

### 3.3. Genome-Wide Association Studies

Twenty-four loci associated with mixograph-related traits were detected. There were nine loci for MPT distributed across chromosomes 1A, 1B, 1D, 2B, 3A, 3D, 4A, 5D and 7D explaining 10.6–42.5% of the phenotypic variances. A single locus affecting MPV was located on chromosome 5D, explaining 13.4–16.8% of the phenotypic variance. Eight loci for MPW on chromosomes 1B (2), 2A, 3B, 4A, 4B, 6A and 7B explained 10.2–15.8% of the phenotypic variance. Twelve loci for MTxW were identified on chromosomes 1A, 1B (2), 1D, 2B, 3D (2), 4A (2), 7B and 7D (2), accounting for 10.5–27.3% of the phenotypic variances. The 1A (506.9 Mb), 1B (553.6 Mb), 1D (407.9–416.5 Mb), 3D (191.1 Mb) and 7D (321.0 Mb) loci showed pleiotropic effects on MPT and MTxW, whereas the 4A locus (610.1–621.6 Mb) controlled both MPW and MPT (Table 2 and Appendix A, Figure 1 and Appendix A).

### 3.4. Candidate Genes

Seven candidate genes were identified for wheat progressing quality-related traits. Three 3-ketoacyl-CoA synthase genes (*TraesCS2B01G535700*, *TraesCS3A01G451100* and *TraesCS3D01G444000*) were identified on chromosomes 2B (731.5 Mb), 3A (689.5 Mb) and 3D (553.2 Mb), and another gene encoding lipoxygenase (*TraesCS4A01G359800*) was detected in the QTL (632.9 Mb) on chromosome 4A. Candidate genes for plant lipid transfer protein/Par allergen (*TraesCS4A01G021300*), pyridoxal phosphate-dependent decarboxylase (*TraesCS1A01G329500*) and sucrose synthase 3 (*TraesCS3D01G184500*) were identified on chromosomes 4A (14.8 Mb), 1A (518.5 Mb) and 3D (170.2 Mb), respectively (Table 3 and Appendix A).

## 4. Discussion

As a vital staple crop and an engine for advancing agricultural high-quality development, particularly with regard to enhancing wheat quality, there exists an increased urgency towards such efforts. Over the past several years, advancements have been achieved by means of genetic modifications within the context of complicated wheat quality characteristics. Nonetheless, owing to the challenges posed by the lack of accurate trait diagnosis methods when it comes to wheat quality, studies on its underlying genetics are moving at a snail’s pace, further impeding the overall wheat quality breeding processes. To address these issues, researchers should explore the highly correlated marker systems linked to various aspects of wheat quality via genealogical analyses so as to determine relevant regions that could be employed later during wheat quality breeding—ultimately speeding up the whole selection procedure.

The rheological properties of dough are an essential quality trait for wheat, determining not only its own processing qualities but also those of breads and cooked products made using it. Currently, various instruments such as the farinograph, rheometer and mixograph are primarily used to assess these traits. However, due to the relatively high quantity of flour needed (normally exceeding 100 g) and the lengthy testing times involved, these devices prove difficult to apply when confronted with numerous samples during breeding generations. In contrast, the mixograph calls for less material and features shorter test times than the aforementioned equipment. Furthermore, it displays highly significant correlations with key parameters measured by means of the farinograph and rheometer, while simultaneously demonstrating a strong connection with actual kneading times during baking. Consequently, this tool is presently extensively implemented in both domestic and international wheat breeding initiatives [13,14].

In this study, most of the 165 accessions mainly from Shandong and foreign cultivars were classified into subgroup 1; subgroup 2 consisted of 54 accessions, mainly from Henan, Anhui and Shaanxi provinces, whereas subgroup 3 mainly included the accessions from Henan province. Thus, the MLM model with population structure and kinship matrix settings was performed in this study to avoid spurious results [22]. The LD decay distance for the whole genome of about 8 Mb indicated that the marker density was adequate for the further association analysis [25].

### 4.1. Comparison with QTLs or Genes in Previous Studies

Dough strength constitutes a prototypical quantitative trait under the influence of multiple minor genes. Numerous QTLs associated with mixograph-related traits have been previously documented [13,14,15,16,17]. However, those studies were mainly based on simple sequence repeat (SSR) markers and not tightly associated with QTLs. In this study, association analysis of mixograph-related traits was performed using high-density SNP arrays. Barakat et al. [2] reported 108 loci for farinograph- and mixograph-related traits on all 21 wheat chromosomes in two double haploid (DH) populations. Three loci on chromosomes 1B, 2B and 3B overlapped present loci on chromosomes 1B (13.5–14.5 Mb) for MTxW, 2B (738.2–752.8 Mb) for MPT and 3B (561.0–579.3 Mb) for MPW. Zhang-Biehn et al. [28] have reported 11 loci for mixograph-related traits on chromosomes 1A, 1B, 1D, 2A, 5B, 6A and 6D in a panel of 462 advanced breeding lines; three overlapped SNPs were present in 1B locus (553.6 Mb) for MPT and MTxW, 1D (407.9–416.5 Mb) for MPT and MTxW, and 6A (23.4–26.9 Mb) for MPW. The locus for MPT and MTxW on chromosome 1D (407.9–416.5 Mb) was in the same position as *Qmlt.tamu.1d* identified by Yu et al. [32], who also reported *Qgcd.tamu.7B*, which overlapped with the present locus for MPW on chromosome 7B (551.9–557.4 Mb). Zhou et al. [33] reported two loci for mixograph-related traits on chromosomes 4A and 7B that coincided with the present 4A (621.2–667.7 Mb) for MPT and 7B (321.0 Mb) loci for MPT and MTxW. However, due to the fact that most genetic analyses of wheat quality traits have been conducted using traditional SSR and DArT markers, which lack specific physical locations and cannot be directly compared with the SNP information employed in this study, we are limited to the following inferences: among the 24 loci for mixograph-related traits, nine were reported previously; the remaining loci may be novel.

### 4.2. Candidate Gene Analysis

Candidate gene *TraesCS1A01G329500* on chromosome 1A encoding a pyridoxal phosphate-dependent decarboxylase involved in nitrogen metabolism affects the amount and composition of dough development and gluten matrix [34]. *TraesCS4A01G021300* on chromosome 4A in an LD decay with a nonspecific lipid transfer protein (LTP) gene promotes the inter-membrane transfer of lipids that plays an important role in dough mixing. Lipids not only influence the dough capacity, but also interact with glutenin and gliadin proteins [35]. *TraesCS3D01G184500* on chromosome 3D encodes sucrose phosphate synthase III (SPS III), which plays an important role in the sucrose metabolic pathway in plants [36,37]. SPS III is an important enzyme involved in the metabolism of sugars in plant cells. Its main function is to convert glucose and fructose into sucrose and, at the same time, produce a phosphate group. Fructose 6-phosphate and UDP glucose are catalyzed by sucrose phosphate synthase to form sucrose 6-phosphate, which is catalyzed by sucrose phosphatase to form sucrose. Within the LD decay of a locus on chromosome 4A (632.9 Mb), there was (*TraesCS4A01G359800*, LOX 2) encoding lipoxygenase 2 that catalyzes bound fatty acids [38,39]. Previous studies showed that lipids form complexes with amylose and LOX 2 acts on the complex formed by starch and lipids to increase the solubility of starch [40]. *TraesCS2B01G535700* on chromosome 2B (731.5 Mb), *TraesCS3A01G451100* on chromosome 3A (689.5 Mb) and *TraesCS3D01G444000* on chromosome 3D (553.2 Mb) encode 3-ketoyl COA synthases involved in the fatty acid biosynthesis pathway [41]. Previous studies indicated that interaction between lipids and proteins has a significant impact on the rheological properties of dough. Further experiments are needed to verify the functions of the candidate genes due to the complexity of metabolic pathways and genetic backgrounds.

While conventional wheat breeding practices have contributed to the enhancement of processing quality, the efficacy of early-generation selection remains limited. Additive effects have been discerned between traits associated with processing quality and advantageous alleles recognized through genome sequence variations. The accumulation of multiple favorable alleles, such as those pinpointed in this investigation, holds the potential to enhance processing quality-linked traits. Loci with stable effects across environments should be more appliable in MAS, such as those on chromosomes1A (506.9 Mb), 1B (553.6 Mb), 1D (407.9–416.5 Mb) and 7D (321.0 Mb), showing pleiotropic effects on MPT and MTxW, along with the 4A (610.1–621.6 Mb) locus with pleiotropic effects on MPW and MPT. The loci that have been confirmed through conventional linkage mapping or GWAS in prior research could serve as subjects for subsequent investigations. Finally, accessions with superior processing quality-related traits and larger numbers of favorable alleles, such as Aca 601, Klein Jaba l1, Shanyou 225, Jishi 02-1, Sagittario, Klein Flecha, Wanmai 33, Sunong6, Libero Mantol, ProINTA Colibr 1 and Nidera Baguette 20, are recommended as parental lines for breeding cultivars with superior quality-related traits by MAS (Table 4).

## 5. Conclusions

A GWAS for mixograph-related traits was conducted on a panel of 165 varieties using the wheat 90K and 660K SNP arrays. In total, 15 of the 24 loci identified were novel. Seven candidate genes for mixograph-related traits were predicted. SNPs in genes associated with favorable mixograph traits can be converted to selection markers that can be used in the early generations of breeding.

## Figures and Tables

**Figure 1 genes-14-01816-f001:**
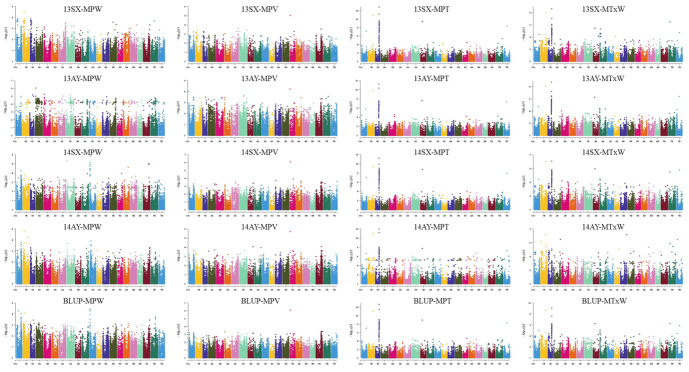
Manhattan plots for mixograph-related traits in 165 wheat accessions by the mixed linear model (MLM) in Tassel v5.0. MPT: mixograph midline peak time; MPV: mixograph midline peak value; MPW: mixograph midline peak width; MTxW: mixograph midline 8 min band width.13SX, 13AY, 14SX, 14AY and BLUP indicate Suixi 2013, Anyang 2013, Suixi 2014, Anyang 2014 and the best linear unbiased prediction (BLUP), respectively.

**Table 1 genes-14-01816-t001:** ANOVA for mixograph-related traits in a panel of 165 wheat accessions.

Source of Variation	df	MS			
		MPW	MPV	MPT	MTxW
Genotypes	164	1.88 **	46.3 **	32.5 **	20.3 **
Environments	3	0.42 **	295.0 **	896.3 **	56.4 **
Replicates (nested in environments)	2	0.15 **	6.1 **	7.1 **	3.2 **
Genotype*Environment	983	0.13 **	3.8 **	5.9 **	2.1 **
Error	1425				

** significant at *p* = 0.01. MPW: mixograph midline peak width; MPV: mixograph midline peak value; MPT: mixograph midline peak time; MTxW: mixograph midline 8 min band width.

**Table 2 genes-14-01816-t002:** Mixograph-related traits identified in the wheat accession panel by association analysis.

Loci	Trait	Chr.	Start (Mb)	R^2^		*p*-Value		Environments	Favorable Allele	Reference
				Min	Max	Min	Max			
qM1	MPT	1A	506.9	14.90%	23.70%	6.70 × 10^−8^	1.90 × 10^−5^	E1; E3; E4; E5	C	
	MTxW	1A	506.9	13.80%	15.90%	9.40 × 10^−6^	3.90 × 10^−5^	E1; E3; E5	T	
qM2	MTxW	1B	13.5–14.5	10.80%	17.00%	7.20 × 10^−7^	6.40 × 10^−5^	E1; E4; E5	T	
qM3	MPW	1B	130.6	10.30%	12.10%	1.70 × 10^−5^	7.40 × 10^−5^	E1; E4; E5	A	[2]
	MPT	1B	553.6	31.50%	37.70%	9.10 × 10^−12^	4.40 × 10^−10^	E1; E3; E4; E5	G	
qM4	MTxW	1B	553.6	14.80%	27.10%	1.90 × 10^−9^	3.70 × 10^−6^	E1; E2; E3; E4; E5	G	
qM5	MPW	1B	673.4–674.5	12.70%	13.90%	3.70 × 10^−5^	8.10 × 10^−5^	E2	A	[28]
qM6	MPT	1D	407.9–416.5	12.60%	42.50%	1.40 × 10^−13^	9.80 × 10^−5^	E1; E2; E3; E4; E5	G	[32]
	MTxW	1D	407.9–416.5	10.80%	27.30%	8.30 × 10^−10^	9.40 × 10^−5^	E1; E2; E3; E4; E5	G	
qM7	MPW	2A	191.9–199.6	12.50%	13.80%	3.50 × 10^−5^	9.00 × 10^−5^	E2	A	[28]
qM8	MTxW	2B	4.6	10.60%	11.20%	6.30 × 10^−5^	9.40 × 10^−5^	E2; E3; E5	G	[32,33]
qM9	MPT	2B	738.2–752.8	14.90%	18.20%	4.70 × 10^−6^	2.70 × 10^−5^	E4	G	[2]
qM10	MPT	3A	709.6–710.7	12.90%	11.60%	4.20 × 10^−5^	9.70 × 10^−5^	E4	G	
qM11	MPW	3B	561.0–579.3	13.00%	13.60%	4.00 × 10^−5^	7.80 × 10^−5^	E2	A	
qM12	MPT	3D	191.1	22.70%	28.90%	3.60 × 10^−10^	1.70 × 10^−8^	E1; E3; E4; E5	A	[33]
	MTxW	3D	191.1	14.70%	17.30%	5.60 × 10^−7^	3.70 × 10^−6^	E1; E2; E3; E5	A	
qM13	MTxW	3D	578.4	10.90%	11.70%	3.10 × 10^−5^	6.90 × 10^−5^	E1; E4; E5	A	
qM14	MTxW	4A	12.4–12.6	12.60%	13.50%	8.20 × 10^−6^	9.50 × 10^−5^	E1; E2; E3; E5	A	
qM15	MTxW	4A	89.9–90.4	10.50%	14.40%	4.60 × 10^−6^	8.10 × 10^−5^	E1; E5	A	
qM16	MPW	4A	610.1–621.6	12.80%	14.40%	2.60 × 10^−5^	9.00 × 10^−5^	E2	G	[2]
	MPT	4A	621.2–667.7	14.60%	17.60%	4.30 × 10^−6^	4.90 × 10^−5^	E1; E4; E5	G	
qM17	MPW	4B	12.9–25.8	10.20%	15.80%	1.30 × 10^−6^	7.70 × 10^−5^	E2; E3; E5	G	
qM18	MPV	5D	3.6	13.40%	16.80%	8.20 × 10^−7^	9.50 × 10^−6^	E1; E3; E4; E5	C	
qM19	MPT	5D	454.1	10.60%	12.30%	2.30 × 10^−5^	8.50 × 10^−5^	E1; E3; E5	G	
qM20	MPW	6A	23.4–26.9	12.70%	14.10%	3.70 × 10^−5^	9.60 × 10^−5^	E2	G	
qM21	MTxW	7B	126.6	14.80%	18.20%	3.20 × 10^−7^	3.20 × 10^−6^	E1; E3; E4; E5	C	
qM22	MPW	7B	551.9–557.4	12.50%	15.20%	2.00 × 10^−5^	9.90 × 10^−5^	E2	G	
qM23	MTxW	7D	57.5	13.90%	22.30%	3.40 × 10^−7^	6.00 × 10^−5^	E2; E4; E5	C	
qM24	MPT	7D	321	20.80%	25.10%	3.70 × 10^−9^	5.90 × 10^−8^	E1; E3; E4; E5	G	
	MTxW	7D	321	12.40%	17.80%	4.00 × 10^−7^	1.90 × 10^−5^	E1; E2; E3; E5	C	[33]

MPW: mixograph midline peak width; MPT: mixograph midline peak time; MPV: mixograph midline peak value; MTxW: mixograph midline 8 min band width. E1, E2, E3, E4 and E5 indicate Suixi 2013, Anyang 2013, Suixi 2014, Anyang 2014 and the best linear unbiased prediction (BLUP), respectively.

**Table 3 genes-14-01816-t003:** Candidate genes for mixograph-related traits.

Candidate Gene	Chromosome	Position (Mb)	Annotation
*TraesCS1A01G329500*	1A	518.5	Pyridoxal phosphate-dependent decarboxylase
*TraesCS2B01G535700*	2B	731.5	3-ketoacyl-CoA synthase
*TraesCS3A01G451100*	3A	689.5	3-ketoacyl-CoA synthase
*TraesCS3D01G444000*	3D	553.2	3-ketoacyl-CoA synthase
*TraesCS3D01G184500*	3D	170.2	sucrose synthase 3
*TraesCS4A01G021300*	4A	14.8	Plant lipid transfer protein/Par allergen
*TraesCS4A01G359800*	4A	632.9	Lipoxygenase

**Table 4 genes-14-01816-t004:** Accessions could be used for wheat processing quality improvement.

Cultivar	BLUP-MPT	BLUP-MPV	BLUP-MPW	BLUP-MTxW
Aca 601	5.6	50.0	26.7	17.7
Klein Jabal 1	6.5	50.4	20.6	17.2
Shanyou 225	4.6	54.0	24.4	16.6
Jishi 02-1	6.0	52.8	22.5	16.0
Sagittario	5.1	50.6	23.4	15.6
Klein Flecha	5.6	49.5	23.4	15.3
Wanmai 33	6.3	51.9	21.5	15.1
Sunong 6	6.2	48.4	18.9	14.8
Libero	5.6	44.2	18.4	14.8
Mantol	5.5	47.3	19.5	14.5
ProINTA Colibr 1	5.2	51.4	21.2	14.4
Nidera Baguette 20	5.0	47.2	21.6	13.4
Barra	6.3	44.0	15.1	13.2
Aca 801	5.4	48.3	23.8	13.0
Jimai 20	4.0	50.6	23.5	12.8
Nidera Baguette 10	5.8	42.4	15.3	12.5
Shanmai 94	4.6	52.9	22.8	11.8
Kitanokaori	4.0	49.4	17.6	11.5
Xinong 979-005	4.3	52.5	25.2	11.4
Norin 67	3.0	54.6	23.5	11.1
Shanmai 509	6.1	44.0	16.4	11.0
Zhoumai 26	5.6	41.6	14.5	10.8
Jinan 17	3.5	52.6	25.7	10.6
Sunstate	4.4	45.3	16.7	10.6
Zhoumai 19	4.3	49.5	19.5	10.3
Gaocheng 8901	4.9	52.4	19.2	10.3
Shannong 981	3.2	63.8	30.4	10.2
Genio	3.5	48.4	23.5	10.1
Shiyou 17	4.5	47.2	17.7	9.9
Jining 16	4.6	49.1	18.8	9.7

## Data Availability

All datasets generated for this study are included in the article/Appendix A; further inquiries can be directed to the first author.

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
