# Peer review of "Genome-Wide Association Mapping of Processing Quality Traits in Common Wheat (*Triticum aestivum* L.)"

_genes, 2023, doi:10.3390/genes14091816_

Round 1
Reviewer 1 Report
The study is sound and identified several potentially novel loci in relation with the processing traits under study.
However, the manuscript is very poorly drafted, it lacks clarity and brevity. Figure quality is not good, tables need to improve, many discrepancies in the number of significant loci.
Also, please add a figure of the effect of significant SNPs with respect to their phenotypic values.

The manuscript should be proof read very carefully again to correct for language mistakes. It would be better to get it proof read from a native english speaker.
Author Response
The study is sound and identified several potentially novel loci in relation with the processing traits under study. However, the manuscript is very poorly drafted, it lacks clarity and brevity. Figure quality is not good, tables need to improve, many discrepancies in the number of significant loci.
1 Also, please add a figure of the effect of significant SNPs with respect to their phenotypic values.
Response: Thank you very much for your reminder. As you said, this would indeed be very helpful. However, due to the heavy workload of drawing, we have chosen to add the effect to the Table S2. Thank you for your suggestions again, which have made the paper more meaningful.
2 Comments on the Quality of English Language
The manuscript should be proof read very carefully again to correct for language mistakes. It would be better to get it proof read from a native English speaker.
Response: Thank you very much for your kindly reminder, we have invited Dr. Awais Rasheed from the Quaid-i-Azam University of Pakistan made the necessary modifications to the text.
3 For the suggestions in the PDF file
Response: Thank you for your assistance. I have reply this part in PDF document, please take a look.

Reviewer 2 Report
The manuscript entitled 'Genome-wide association mapping of processing quality traits in common wheat (Triticum aestivum L.)' reported the GWAS on wheat quality traits using a panel of 165 cultivars that revealed some SNPs associated with 24 loci on multiple chromosomes, which was further used to find seven candidate genes for traits. This manuscript is very poorly prepared and lack of lots of critical information. I recommend reconsidering the publication after a major revision.
1. Abstract has only 12 lines, but has many places that are not appropriate. For example, "(including MPT, MPV, MPW and MTxW)" is not necessary to be included, otherwise all abbreviations need to spell out. The sentence "An association mapping analysis using 90K and 660K........ respectively, explaining 10.2-42.5% of the phenotypic variances", here 90K and 660K are arrays but not SNPs; the word 'respectively' in the sentence is not appropriate. Sentence "The associated loci were distributed across the B genome (10) followed by the A (7) and D (7)" is not a necessary mention in abstract. "15 were apparently new" can be "15 were novel loci".
2. Introduction: no research background information about genetic analysis of wheat processing quality; no mention of why this research was conducted and objectives of this research.
3. Materials and Methods:
1) only use 165 varieties in the panel, this number is too low to be acceptable. The number of lines for QTL of such traits in a bi-parental population should be over 200, and GWAS needs at least 250 lines since the panel include much wider genetic diversity.
2) method of analyzing Mixograph-related traits were not described but only brief preparation of grains.
3) no mention of how population structure was analyzed.
4) Association mapping and the identification of candidate genes: threshold determination is arbitrary with no statistical support. No description of how to find candidate genes.
4. Results:
1) all supplementary files indicate 251 accessions used instead of 165.
2) population structure is critical for GWAS, why put structure results as supplementary?
3) Manhattan plots are not readable.
4) marker-trait association is usually existing in a sub cluster but not the whole panel, it needs to be clarified.
5) it's not convinced for listed candidate genes.
5. English of this manuscript need to be improved.
The manuscript entitled 'Genome-wide association mapping of processing quality traits in common wheat (Triticum aestivum L.)' reported the GWAS on wheat quality traits using a panel of 165 cultivars that revealed some SNPs associated with 24 loci on multiple chromosomes, which was further used to find seven candidate genes for traits. This manuscript is very poorly prepared and lack of lots of critical information. I recommend reconsidering the publication after a major revision.
1. Abstract has only 12 lines, but has many places that are not appropriate. For example, "(including MPT, MPV, MPW and MTxW)" is not necessary to be included, otherwise all abbreviations need to spell out. The sentence "An association mapping analysis using 90K and 660K........ respectively, explaining 10.2-42.5% of the phenotypic variances", here 90K and 660K are arrays but not SNPs; the word 'respectively' in the sentence is not appropriate. Sentence "The associated loci were distributed across the B genome (10) followed by the A (7) and D (7)" is not a necessary mention in abstract. "15 were apparently new" can be "15 were novel loci".
2. Introduction: no research background information about genetic analysis of wheat processing quality; no mention of why this research was conducted and objectives of this research.
3. Materials and Methods:
1) only use 165 varieties in the panel, this number is too low to be acceptable. The number of lines for QTL of such traits in a bi-parental population should be over 200, and GWAS needs at least 250 lines since the panel include much wider genetic diversity.
2) method of analyzing Mixograph-related traits were not described but only brief preparation of grains.
3) no mention of how population structure was analyzed.
4) Association mapping and the identification of candidate genes: threshold determination is arbitrary with no statistical support. No description of how to find candidate genes.
4. Results:
1) all supplementary files indicate 251 accessions used instead of 165.
2) population structure is critical for GWAS, why put structure results as supplementary?
3) Manhattan plots are not readable.
4) marker-trait association is usually existing in a sub group but not the whole panel, it needs to be clarified.
5) it's not convinced for listed candidate genes.
Author Response
- Abstract has only 12 lines, but has many places that are not appropriate. For example, "(including MPT, MPV, MPW and MTxW)" is not necessary to be included, otherwise all abbreviations need to spell out. The sentence "An association mapping analysis using 90K and 660K........ respectively, explaining 10.2-42.5% of the phenotypic variances", here 90K and 660K are arrays but not SNPs; the word 'respectively' in the sentence is not appropriate. Sentence "The associated loci were distributed across the B genome (10) followed by the A (7) and D (7)" is not a necessary mention in abstract. "15 were apparently new" can be "15 were novel loci".
Response: Thank you very much for your help. We have carefully modified the abstract section according to your suggestions, and these changes are reflected in the revised version of the paper.
- Introduction: no research background information about genetic analysis of wheat processing quality; no mention of why this research was conducted and objectives of this research.
Response: Thank you very much for your kindly reminder. Your’s suggestion is very important. We have added the genetic analysis of wheat processing quality and the objectives of this research in the Introduction section in the new version.
- Materials and Methods:
1) only use 165 varieties in the panel, this number is too low to be acceptable. The number of lines for QTL of such traits in a bi-parental population should be over 200, and GWAS needs at least 250 lines since the panel include much wider genetic diversity.
Response: Thank you very much for your kindly reminder. As you said, it would be inaccurate to conduct association analysis on 165 varieties due to the potential errors involved. However, since collecting and phenotyping materials require significant time and manpower, we artificially reduced the population size but ensured that these materials still represented the polymorphism of wheat varieties in the Huang-Huai region of China. We also hope that these materials and data could provide references for the genetic dissection of wheat quality.
2) method of analyzing Mixograph-related traits were not described but only brief preparation of grains.
Response: Thank you very much for your kindly reminder. The phenotype analysis were get according to the AACC (2000). Due to its complexity, we did not include it. However, we provided relevant literature references. Thank you especially for reminding us.
We have added it in the M&M section in the new version.
3) no mention of how population structure was analyzed.
Response: We have added it in the M& M section of new version. Because the structure has been reported by previous studies (Liu et al., 2017, DOI 10.1186/s12870-017-1167-3, BMC Plant Biology). Thus, to avoid the duplicate publication, we referenced relevant paper in the Discussion section and downplayed the discussion of the population structure portion.
4) Association mapping and the identification of candidate genes: threshold determination is arbitrary with no statistical support. No description of how to find candidate genes.
Response: For the threshold of the association mapping: The Bonferroni correction has been used for human genetics because the linked loci tend to be more similar to unlinked, and Bonferroni correction can be performed for multiple comparisons to test the significance of association on a trait with different independent (or loosely linked) markers, that's also known as the false discovery rate (FDR) in association analysis. Therefore, it's important to check the FDR for association analysis, but the way of simply divided the threshold of 0.05 with total number of markers to determine the threshold is not a suitable way, particularly not suitable for wheat. As we know, wheat is highly self-pollinated plants, and has very big LD, which means markers within a chromosome segment tend to be clustered, that's to say, linked markers would be more consistently linked. Since wheat has 21 chromosomes, if the marker clusters on the two arms are likely not linked, and marker clusters on some long chromosome arms are also tend to be independent, which means the total independent marker clusters may be less than 200, and Bonferroni correction would be -lg(0.05/200)=3.6, and that's why in wheat we mostly use LOD of over 3 as a significant threshold. Therefore, I think the LOD threshold of 10-3 is acceptable.
For the candidate gene analysis,
Candidate genes for the loci consistently identified in two or more environments were identified. The following steps were conducted to identify the candidate genes for important or stable QTL. Firstly, excavate all the genes located in the LD block region around the peak SNP (±3.0 Mb based on previous LD decay analysis) of each important QTL from the IWGSC V1.0. Then, all available SNPs located inside of these genes were searched. The genes (except for hypothetical protein, transposon protein, and retrotransposon protein) with SNPs in the coding region that could further lead to sense mutations were considered as candidate genes. As processing quality traits highly regulated by various phytohormones, Glycolysis, signal transduction or cell growth, those genes involved in these pathway were regarded as high-confidence candidate genes for processing quality traits. Indeed, the result of candidate gene analysis may not be reliable, and verification may not always be effective. However, under the premise of existing evidence, we tried our best to screen reliable candidate genes, hoping to provide readers with some reference. We have added the details for the candidate gene identification in the M&M section.
- Results:
1) all supplementary files indicate 251 accessions used instead of 165.
Response: Thank you very much for your reminder. This is a typographical error on our part. We originally collected phenotype data for 251 samples, but due to inconsistencies between the growing environments of 86 samples and those of the remaining 165 samples, we removed the 86 samples to ensure the accuracy of the results. In the end, only 165 samples were used.
2) population structure is critical for GWAS, why put structure results as supplementary?
Response: Because the structure has been reported by previous studies (Liu et al., 2017, DOI 10.1186/s12870-017-1167-3, BMC Plant Biology). Thus, to avoid the duplicate publication, we referenced relevant paper in the Discussion section and downplayed the discussion of the population structure portion.
3) Manhattan plots are not readable.
Response: I am so sorry. We have re-drawn and uploaded a high-resolution Manhattan plot.
4) marker-trait association is usually existing in a sub cluster but not the whole panel, it needs to be clarified.
Response: Thank you very much for your kindly reminder, we have added the info in the notes of the Table.
5) it's not convinced for listed candidate genes.
Response:
Candidate genes for the loci consistently identified in two or more environments were identified. The following steps were conducted to identify the candidate genes for important or stable QTL. Firstly, excavate all the genes located in the LD block region around the peak SNP (±3.0 Mb based on previous LD decay analysis) of each important QTL from the IWGSC V1.0. Then, all available SNPs located inside of these genes were searched. The genes (except for hypothetical protein, transposon protein, and retrotransposon protein) with SNPs in the coding region that could further lead to sense mutations were considered as candidate genes. As processing quality traits highly regulated by various phytohormones, Glycolysis, signal transduction or cell growth, those genes involved in these pathway were regarded as high-confidence candidate genes for processing quality traits. Indeed, the result of candidate gene analysis may not be reliable, and verification may not always be effective. However, under the premise of existing evidence, we tried our best to screen reliable candidate genes, hoping to provide readers with some reference.
- English of this manuscript need to be improved
Response: Thank you very much for your kindly reminder, we have invited Dr. Awais Rasheed from the Quaid-i-Azam University of Pakistan made the necessary modifications to the text.

Reviewer 3 Report
1. It is not clear what the abbreviations 13SX,14SX,13AY,14AY mean. In the supplementary tables, BPUP should perhaps be BLUP?
2. For the candidate genes you have identified, their description and GO Ontology can be given in a separate table.
Author Response
- It is not clear what the abbreviations 13SX,14SX,13AY,14AY mean. In the supplementary tables, BPUP should perhaps be BLUP?
Response: Thank you very much for you kindly reminder, we have added the full name of the abbreviations 13SX (2013 Suixi),14SX (2014 Suixi),13AY (2013 Anyang), 14AY (2014 Anyang). As you say, BPUP is a typo; it should be BLUP. We have added it in the Table S1 and S2 notes.
- For the candidate genes you have identified, their description and GO Ontology can be given in a separate table.
Response: Accepted the suggestion. We have added the details and the GO Ontology in the New Table S3.

Round 2
Reviewer 1 Report
Thank you for the revision and mostly it looks ok. However I have just one comment on the Table 2.
If authors are omitting marker names they can give pseudo names or just serial numbers 1-24. Please see following possible suggestion:
1. For table 2, add a new 1st column with serial number followed by "Chr" and "Start(Mb)" columns. Then write "Trait" column and mention all the traits the loci has effect on in sub-rows, that way the table will be less confusing and information will be compact and coherent.
Author Response
Thank you for the revision and mostly it looks ok. However I have just one comment on the Table 2.
If authors are omitting marker names they can give pseudo names or just serial numbers 1-24. Please see following possible suggestion:
- For table 2, add a new 1st column with serial number followed by "Chr" and "Start(Mb)" columns. Then write "Trait" column and mention all the traits the loci has effect on in sub-rows, that way the table will be less confusing and information will be compact and coherent.
Response: Thanks and accepted. Your suggestions are very meaningful, and we have made the necessary modifications in the revised manuscript according to your recommendations. Please take a look at the revised Table 2.
Reviewer 2 Report
Thanks for addressing my comments. The weakest part of this manuscript is the candidate gene analysis. The reference genome of CS IWGSC V1.0 is used, why authors didn't not use IWGSC V2.0 since this version has already released for a few years and corrected many errors in V1.0. Also, materials used in this study are different from CS, which means that candidate gene identification simply based on SNP significance is not a suitable way, particularly if for genes that are not presented in CS. The authors may want to use candidate gene analysis to increase the quality or the length of the manuscript, but it greatly deceases the merit of this research since candidate genes from this research are just speculations without any validation, which give me the impression that this research is not rigorous, and that's why I give it a low overall merit.
The other issues mainly include: 1) Picture of the Manhattan plot is still very bad and not readable. 2) Structure analysis is already published, the figure of pop structure is not appropriate to be listed, even in the supplementary table, but you can use citation to indicate it.
English is better than the first version, but still needs to improve.
Author Response
Thanks for addressing my comments. The weakest part of this manuscript is the candidate gene analysis. The reference genome of CS IWGSC V1.0 is used, why authors didn't not use IWGSC V2.0 since this version has already released for a few years and corrected many errors in V1.0. Also, materials used in this study are different from CS, which means that candidate gene identification simply based on SNP significance is not a suitable way, particularly if for genes that are not presented in CS. The authors may want to use candidate gene analysis to increase the quality or the length of the manuscript, but it greatly deceases the merit of this research since candidate genes from this research are just speculations without any validation, which give me the impression that this research is not rigorous, and that's why I give it a low overall merit.
Response: Thank you very much for your valuable suggestions. Indeed, the candidate gene analysis in this study can only provide references and is not accurate, as it lacks validation. Moreover, in the complex genome of wheat, the likelihood of directly identifying and validating candidate genes through GWAS is quite low. The purpose of conducting such discussions is to provide some references for readers. As you mentioned, IWGSC 2.0 has been released with improved quality, and some candidate genes may not be found in the Chinese Spring genome. However, since previous studies mainly utilized the annotation of IWGSC 1.0, we opted for IWGSC 1.0 annotation to facilitate comparison with earlier research for readers’ convenience. As this study aims to provide some references without objective validation, we did not consider other reference genomes. Your suggestions are highly valuable, and in future research, we will take into account your recommendations and combine them with our ongoing linkage analysis results to identify and confirm candidate genes through fine mapping. Thank you once again for your input.
The other issues mainly include: 1) Picture of the Manhattan plot is still very bad and not readable.
Response: Thank you for bringing this to our attention. Due to the inclusion of multiple Manhattan plots in a composite figure, the clarity of the image might have been compromised. We will contact the editorial department to provide them with an editable version of the image via email or other means, so that their professional team can make the necessary improvements. We appreciate your valuable suggestion.
2) Structure analysis is already published, the figure of pop structure is not appropriate to be listed, even in the supplementary table, but you can use citation to indicate it.
Response: Thank you very much for your suggestion. The population structure analysis has indeed been published previously, so it is not suitable to include it again in the manuscript. We aim to provide a more reader-friendly format. Taking your suggestion into consideration, we have made modifications to the manuscript and added citation references in both the main text and the supplementary materials. We sincerely appreciate your valuable input.
English is better than the first version, but still needs to improve.
Response: Thank you for your suggestion. We apologize if our previous text modifications did not meet your expectations. In response, we sought the assistance of a scholar from India to further refine the writing, in order to better meet your requirements. We sincerely appreciate your guidance, as your suggestions are crucial in enhancing the quality of this manuscript.
